# TOWARDS UNDERSTANDING ROBUST MEMORIZATION IN ADVERSARIAL TRAINING

## ABSTRACT

Adversarial training is a standard method to train neural networks to be robust to adversarial perturbation. However, in contrast with benign overfitting in the standard deep learning setting, which means that over-parameterized neural networks surprisingly generalize well for unseen data, while adversarial training method is able to achieve low robust training error, there still exists a significant robust generalization gap, which promotes us exploring what mechanism leads to robust overfitting during learning process. In this paper, we propose an implicit bias called *robust memorization* in adversarial training under the realistic data assumption. By function approximation theory, we prove that ReLU nets with efficient size have the ability to achieve robust memorization, while robust generalization requires exponentially large models. Then, we demonstrate robust memorization in adversarial training from both empirical and theoretical perspectives. In particular, we empirically investigate the dynamics of loss landscape over input, and we also provide theoretical analysis of robust memorization on data with linear separable assumption. Finally, we prove novel generalization bounds based on robust memorization, which further explains why deep neural networks have both high clean test accuracy and robust overfitting at the same time.

## 1 INTRODUCTION

Although deep learning has made a remarkable success in many application fields, such as computer vision (Voulodimos et al., 2018) and natural language process (Devlin et al., 2018), it is well-known that only small but adversarial perturbations additional to the natural data can make well-trained classifiers confused (Szegedy et al., 2013; Goodfellow et al., 2014), which promotes designing adversarial robust learning algorithms. In practice, adversarial training methods (Madry et al., 2017; Shafahi et al., 2019; Zhang et al., 2019) are widely used to improve the robustness of models by regarding perturbed data as training data. However, while these robust learning algorithms are able to achieve high robust training accuracy, it still leads to a non-negligible robust generalization gap (Raghunathan et al., 2019), which is also called *robust overfitting* (Rice et al., 2020; Yu et al., 2022).

To explain this puzzling phenomenon, a series of works have attempted to provide theoretical understandings from different perspectives. Despite these aforementioned works seem to provide a series of convincing evidence from theoretical views in different settings, there still exists a gap between theory and practice for at least two reasons.

*First*, although previous works have shown that adversarial robust generalization requires more data and larger models (Schmidt et al., 2018; Li et al., 2022), it is unclear that what mechanism, during adversarial training process, directly causes robust overfitting. In other words, we know there is no robust generalization gap for a trivial model that only guesses labels totally randomly (e.g. deep neural networks with random initialization), which implies that we should take learning process into consideration to analyze robust generalization.

*Second and most importantly*, while some works (Tsipras et al., 2018; Zhang et al., 2019) point out that achieving robustness may hurt clean test accuracy, in most of the cases, it is observed that drop of robust test accuracy is much higher than drop of clean test accuracy in adversarial training (Schmidt et al., 2018; Raghunathan et al., 2019) (see in Table 1). Namely, a weak version of benign overfitting, which means that overparameterized deep neural networks can both fit random data powerfully and generalize well for unseen clean data, remains after adversarial training.

Table 1: Train and test performances on CIFAR-10 (Raghunathan et al., 2019)

|  | Clean training | Adversarial training |
|---|---|---|
| Robust test | 3.5% | 45.8% |
| Robust train | — | 100% |
| Clean test | 95.2% | 87.3% |
| Clean train | 100% | 100% |

Therefore, it is natural to ask the following question:

*What happens, during adversarial learning process, resulting in both benign clean overfitting and significant robust overfitting at the same time?*

In this paper, we provide a theoretical understanding of the adversarial training process by proposing a novel implicit bias called *robust memorization* under the realistic data assumption, which explains why adversarial training leads to both high clean test accuracy and robust generalization gap.

The fundamental data structure that we leverage is that data can be clean separated by a neural network $f_{\text{clean}}$ with moderate size but this neural classifier is non-robust on data with small margin, which is consistent with the common practice that well-trained neural networks have good clean performance but fail to classify perturbed data due to the close distance between small margin data and decision boundary. The existence of small and large margin data for image classification has been empirically verified (Banburski et al., 2021). And we also assume that data is well-separated, which means that there exists a robust classifier in general (Yang et al., 2020). However, this robust classifier may be hard to approximate by neural networks (Li et al., 2022). In other words, clean training often finds simple but non-robust solution, although robust classifier always exists.

Specifically, we consider the underlying data distribution $\mathcal{D}$ where $\mu$ proportional data is with small margin, which can be perturbed as adversarial example, and $1 - \mu$ proportional data is with large margin, which can be robustly classified by the neural network $f_{\text{clean}}$. In adversarial training, we access $N$ instances $\mathcal{S}$ randomly drawn from $\mathcal{D}$ as training data. In fact, by applying concentration technique in high dimensional probability, we know that there roughly exist $\mu N$ small margin training data and $(1 - \mu)N$ large margin training data.

In order to answer the question that we ask, under the above realistic data assumption, we consider the following classifier,

$$f_{\text{adv}}(x) := \underbrace{\sum_{(x_i, y_i) \in S_{\text{small}}} y_i \mathbb{I}\{\|x - x_i\|_p \leq \delta\}}_{\text{robust local indicators on small margin data}} + \underbrace{f_{\text{clean}}(x) \mathbb{I}\left\{\min_{(x_i, y_i) \in S_{\text{small}}} \|x - x_i\|_p > \delta\right\}}_{\text{clean classifier on other data}},$$

where $\mathcal{S}_{\text{small}}$ is the small margin part of training dataset, and $\delta$ denotes adversarial perturbation radius, which is smaller than the separation between data in our setting.

Indeed, the classifer $f_{\text{adv}}$ satisfies all main characteristics of deep models outputted by adversarial training (Proposition 3.2): First, adversarial training error with respect to $f_{\text{adv}}$ achieves zero; second, $f_{\text{adv}}$ has a good clean performance because the classifier is robust within the neighborhood of training data with small margin and it performs as the same as the clean classifier on other data; third, although this classifier has low robust training error, it is non-robust on other data since it only robustly memorizes training data with small margin, which results in robust overfitting.

Inspired by the ideal model $f_{\text{adv}}$, we then propose a conjecture that deep neural networks converge to solution similar to $f_{\text{adv}}$ in adversarial training, which is a novel implicit bias of adversarial training, and we call it *robust memorization*, which provides a theoretical understanding of adversarial training, including why solution found by adversarial training has both good clean generalization and robust generalization gap.

Based on function approximation theory, we can prove that, for $d-$dimensional training dataset $\mathcal{S}$ with $N$ samples, ReLU networks with $\tilde{O}(\mu N d + \text{poly}(d))$ parameters are able to represent the target functions $f_{\text{adv}}$ that we mention for robust memorization (Theorem 3.3 and Corollary 3.4). However, we still prove a lower bound for the network size that is exponential in the data dimension

$d$ to make robust generalization (Theorem 3.5), and it manifests that robust classifier has higher representation complexity than the classifier with robust memorization, which potentially explain why adversarial training converges to robust memorization regime rather than finds robust classifier due to the inductive bias of low representation complexity.

To further demonstrate robust memorization in adversarial training, we empirically investigate the dynamics of loss landscape over input during adversarial learning process. Concretely, for the loss function $\mathcal{L}(f(x), y)$ over classifier $f$ and on data point $(x, y)$, we verify whether $\hat{f}$ outputted by adversarial training is with robust memorization by certain measure — maximum gradient norm within the neighborhood of training data input,

$$\frac{1}{N} \sum_{i=1}^{N} \max_{\|\xi\|_p \leq \delta} \|\nabla_x \mathcal{L}(f(x_i + \xi), y_i)\|_q,$$

where $\{(x_i, y_i)\}_{i=1}^{N}$ denotes the training dataset, and $\frac{1}{p} + \frac{1}{q} = 1$.

This measure reflects the local flatness of loss landscape over input on training dataset. In robust memorization regime, the loss landscape will be very flat within the adversarial training perturbation radius but very sharp outside the neighborhood. Through numerical experiments, the dynamics of the measure observed in different perturbation radius $\delta$ and different training epoch shows that the behavior of models trained by adversarial training is very similar to robust memorization. More details are presented in Section 4.

To further support our conjecture, we theoretically analyze the optimization process of adversarial training under a simple linear separable data set. Our results (Theorem 5.1 and Theorem 5.2) show that clean classification on large margin data implies good clean performance on small margin data, and ReLU networks with moderate size can robustly memorize small margin data efficiently. In other words, we prove that after adversarial training, the network exactly learns the function we described in Section 3.

Finally, motivated by robust memorization, we propose clean and robust generalization bounds. On one hand, clean generalization bound (Theorem 6.2) shows that clean sample complexity is polynomial in data dimension $d$. On the other hand, we derive a upper bound of robust generalization gap (Theorem 6.3) that only relies on the number of samples and global flatness of loss landscape over input. Since adversarial training only promotes model robustly memorizing training data, the global flatness of landscape has no guarantee, and it is also verified by numerical results, which explains why robust generalization gap is large although robust training error can be very low in adversarial training,

## 2 RELATED WORK

**Adversarial Examples and Adversarial Training (AT).** Szegedy et al. (2013) first made an intriguing discovery: even state-of-the-art deep neural networks are vulnerable to adversarial examples. Then, Goodfellow et al. (2014) proposes FGSM to seek adversarial examples, and it also introduce the adversarial training framework to enhance the defence of models against adversarial perturbations. Madry et al. (2017) designs PGD to make adversarial attack and uses it to improve adversarial training, which can be viewed as the multi-step vision of FGSM. In general, FGSM-based AT and PGD-based AT are commonly regarded as the standard methods for adversarial training.

**Robust Generalization Gap (Robust Overfitting).** One surprising behavior of deep learning is that over-parameterized neural networks can generalize well, which is also called *benign overfitting* that deep models have not only the powerful memorization but a good performance for unseen data (Zhang et al., 2017; Belkin et al., 2019). However, in contrast to the standard (non-robust) generalization, for the robust setting, Rice et al. (2020) empirically investigates robust performance of models based on adversarial training methods, which are designed to improve adversarial robustness (Szegedy et al., 2013; Madry et al., 2017), and the work shows that *robust overfitting* can be observed on multiple datasets.

**Theoretical Understanding of Robust Overfitting.** Schmidt et al. (2018); Balaji et al. (2019); Dan et al. (2020) study the *sample complexity* for adversarial robustness, and their works manifest that adversarial robust generalization requires more data than the standard setting, which gives

an explanation of the robust generalization gap from the perspective of statistical learning theory. And another line of works (Tsipras et al., 2018; Zhang et al., 2019) propose a principle called the *robustness-accuracy trade-off* and have theoretically proven the principle in different setting, which mainly explains the widely observed drop of robust test accuracy due to the trade-off between adversarial robustness and clean test accuracy. Recently, Li et al. (2022) investigates the robust expressive ability of neural networks, which demonstrates that, for the well-separated dataset, robust generalization requires exponentially large models, so the hardness of robust generalization may stem from the *expressive power* of practical models.

**Memorization in Adversarial Training.** Dong et al. (2021); Xu et al. (2021) empirically and theoretically explore the memorization effect in adversarial training for promoting a deeper understanding of model capacity, convergence, generalization, and especially robust overfitting of the adversarially trained models. However, different from their works, the concept *robust memorization* proposed in our paper focuses on both robust overfitting and high clean test accuracy, which means that there is surprisingly no clean memorization or clean overfitting.

## 3 ROBUST MEMORIZATION IN ADVERSARIAL TRAINING

In this section, we first introduce some preliminaries in our theoretical framework.

We consider a binary classification task $\mathcal{X} \rightarrow \mathcal{Y}$, where we use $\mathcal{X} \in [0,1]^d, \mathcal{Y} = \{-1, 1\}$ to denote the supporting set of all data input and ground-truth labels, respectively, and the data input dimension is $d$. Let $\mathcal{D}$ be the underlying data distribution. We use clean $0 - 1$ loss to evaluate clean performance of classifier as $\mathcal{L}_{\mathcal{D}}^{\text{clean}}(f) := \mathbb{E}_{(x,y)\sim\mathcal{D}}[\mathbb{I}\{\text{sgn}(f(x)) \neq y\}]$.

Then, we assume that data can be clean separated by a ReLU network with reasonable size (width and depth). Specifically, we assume that there exists a ReLU network classifier $f_{\text{clean}}$ such that $\mathcal{L}_{\mathcal{D}}^{\text{clean}}(f_{\text{clean}}) = 0$, where the clean classifier is defined as

$$f_{\text{clean}}(x) := W_L\sigma(W_{L-1}\sigma(\ldots\sigma(W_1x + b_1)\ldots) + b_{L-1}) + b_L,$$

where $W_i \in \mathbb{R}^{m_i \times m_{i-1}}, b_i \in \mathbb{R}^{m_i}, 1 \leq i \leq L, m_0 = d, m_L = 1$, and we use $\sigma(\cdot)$ to denote ReLU activation function, which is defined as $\sigma(\cdot) = \max\{0, \cdot\}$. Besides, we consider that the width $\max\{m_0, m_1, \cdots, m_L\}$ is $O(d)$, and the depth $L$ is constant.

To understand adversarial robustness by a geometry way, we then introduce a notion called *decision boundary*. Formally, the decision boundary of a classifier $f$ is defined as,

$$DB(f) := \{x \in \mathcal{X} \mid f(x) = 0, \forall\epsilon > 0, \exists x', x'' \in B_p(x, \epsilon), s.t.f(x')f(x'') < 0\}.$$

Notice that this definition is different from that in Zhang et al. (2019), since it requires that the sign of the neighborhood of the decision boundary can be changed, which helps us to establish the relative between decision boundary and adversarial robust margin. We define $l_p$ adversarial margin over classifier $f$ and data point $(x, y)$ as $\min_{\|\xi\|_p \leq \delta} yf(x + \xi)$, then we have the following proposition.

**Proposition 3.1.** *(The equivalent between adversarial margin and distance from decision boundary) Assume that the distance between data point and decision boundary is well-defined, then it holds that,*

$$\{x \in \mathcal{X} \mid \min_{\|\xi\|_p \leq \delta} yf(x + \xi) \geq 0\} = \{x \in \mathcal{X} \mid \text{dist}_p(x, DB(f)) \geq \delta\},$$

$$\{x \in \mathcal{X} \mid \min_{\|\xi\|_p \leq \delta} yf(x + \xi) < 0\} = \{x \in \mathcal{X} \mid \text{dist}_p(x, DB(f)) < \delta\},$$

*where we use* $\text{dist}_p(\cdot, \mathcal{C})$ *to denote* $l_p$ *distance between point* $\cdot$ *and curve* $\mathcal{C}$.

Proposition 3.1 shows that the classifier is robust on data with large distance from decision boundary, and is non-robust on data with small distance from decision boundary. Thus, according distance from decision boundary, we can divide all data into two classes, large margin data and small margin data. Namely, the former is defined as $\mathcal{X}_{\text{large}} = \{x \in \mathcal{X} \mid \text{dist}_p(x, DB(f_{\text{clean}})) \geq \delta\}$, and the latter is defined as $\mathcal{X}_{\text{small}} = \{x \in \mathcal{X} \mid \text{dist}_p(x, DB(f_{\text{clean}})) < \delta\}$. We also consider $\mathbb{P}_{(x,y)\sim\mathcal{D}}\{x \in \mathcal{X}_{\text{small}}\} = \mu$ and $\mathbb{P}_{(x,y)\sim\mathcal{D}}\{x \in \mathcal{X}_{\text{large}}\} = 1 - \mu$, where $0 < \mu < 1$ is a proportional constant.

Another key notion of data in our setting is well-separated, which means that data is far from each other although there exists small margin data. This property is widely observed in Yang et al.

(2020), and it is clear that this assumption is foundation of robust classifier. Formally, we consider the supporting set $\mathcal{X} = A \cup B \subset [0,1]^d$, where two disjoint sets $A, B$ denote positive class and negative class, respectively. And we assume that $\text{dist}_p(A, B) > R$, where we use $\text{dist}_p(\cdot, \cdot)$ to denote $l_p$ distance between two sets.

## 3.1 ROBUST MEMORIZATION UNDER THE REALISTIC DATA ASSUMPTION

In this subsection, we consider adversarial training with access to $N-$sample training dataset $\mathcal{S} = \{(x_1, y_1), (x_2, y_2), \cdots, (x_N, y_N)\}$ i.i.d drawn from the data distribution $\mathcal{D}$, we minimize $0 - 1$ adversarial robust training loss as $\mathcal{L}_{\mathcal{S}}^{\text{adv},\delta}(f) := \frac{1}{N} \sum_{i=1}^{N} \max_{\|\xi\|_p \leq \delta} \mathbb{I}\{\text{sgn}(f(x_i + \xi)) \neq y_i\}$,where $\delta$ is the perturbation radius.

Indeed, we can also divide $\mathcal{S}$ into two sets, small margin training dataset and large margin training dataset. Specifically, let $\mathcal{S}_{\text{small}}$ be the set $\{(x_i, y_i) \in \mathcal{S} \mid \text{dist}_p(x_i, DB(f_{\text{clean}})) < \delta\}$, and $\mathcal{S}_{\text{large}}$ be the set $\{(x_i, y_i) \in \mathcal{S} \mid \text{dist}_p(x_i, DB(f_{\text{clean}})) \geq \delta\}$.

To evaluate robust performance of models, we use $0 - 1$ adversarial robust test loss as $\mathcal{L}_{\mathcal{D}}^{\text{adv},\delta}(f) := \mathbb{E}_{(x,y)\sim\mathcal{D}} \left[ \max_{\|\xi\|_p \leq \delta} \mathbb{I}\{\text{sgn}(f(x + \xi)) \neq y\} \right]$. The final goal of adversarial training is to find good solution with low robust test loss.

Now, we present the concept that we mainly discuss in this work, *robust memorization*. Concretely, we consider the following classifier

$$f_{\text{adv}}(x) := \sum_{(x_i, y_i) \in S_{\text{small}}} y_i \mathbb{I}\{\|x - x_i\|_p \leq \delta\} + f_{\text{clean}}(x) \mathbb{I}\left\{ \min_{(x_i, y_i) \in S_{\text{small}}} \|x - x_i\|_p > \delta \right\}.$$

Under the above realistic data assumption, we can derive the following result.

**Proposition 3.2.** *Assume that data within the $l_p$ $\delta-$ neighborhood of training data has local constant label (which means data within neighborhood of a certain data has the same label), then the classifier $f_{adv}$ has the following properties:*

- *For adversarial robust training error, $\mathcal{L}_{\mathcal{S}}^{adv,\delta}(f_{adv}) = 0$;*

- *For clean test error, $\mathcal{L}_{\mathcal{D}}^{clean}(f_{adv}) = 0$;*

- *For adversarial robust test error, $\mathcal{L}_{\mathcal{D}}^{adv,\delta}(f_{adv}) = \Omega\left(\mathcal{L}_{\mathcal{D}}^{adv,\delta}(f_{clean})\right)$.*

This proposition shows that the classifier $f_{\text{adv}}$ has the same behavior as deep models trained by adversarial training. They have the common performance that they achieve low robust training error and high clean test accuracy, but fail to robust classify the data population $\mathcal{D}$.

Motivated by this, we conjecture that adversarial training can learn functions that are close to this form, which is called *robust memorization*, an implicit bias of adversarial training. In the later sections, we formally analyze the properties of this function and adversarial training systematically from different views, including representation complexity, empirical evidence and theoretical evidence. Finally, based on robust memorization, we establish generalization guarantees to explain good clean generalization and robust generalization gap.

## 3.2 ON THE REPRESENTATION COMPLEXITY OF ROBUST MEMORIZATION

Next, we study the representation complexity of robust memorization, which asks how many parameters are enough to express the robust memorization classifier $f_{\text{adv}}$ via deep neural networks. In fact, only $\tilde{\mathcal{O}}(\mu N d + \text{poly}(d))$ weights are sufficient for ReLU nets to uniformly approximate the target function $f_{\text{adv}}$, which is stated as the following theorem.

**Theorem 3.3.** *Assume that the perturbation radius satisfies $\delta < R/2$, then there exists a classifier $\hat{f}$ represented by a ReLU net with at most*

$$\tilde{O}(\mu N d + \text{poly}(d))$$

*parameters such that $|\hat{f} - f_{adv}| < \epsilon$ for all $x \in [0,1]^d$.*

*Proof sketch of Theorem 3.3* Although an exponentially large number of parameters is *necessary* to approximate a smooth function in general (DeVore et al., 1989), some simple functions can be approximate by neural networks more efficiently Telgarsky (2017). By leveraging this benefit, we use ReLU nets with few parameters to approximate the distance function $d_i(x) := \|x - x_i\|_p$, and we notice that the exact indicator $\mathbb{I}(\cdot)$ can be approximated by a soft indicator with two ReLU neurons. Combined with these results, we can derive Theorem 3.3.

By applying the result of Theorem 3.3, we have the following corollary, and it manifests that a neural network with $\tilde{\mathcal{O}}(\mu N d + \text{poly}(d))$ weights can achieve robust memorization.

**Corollary 3.4.** *Assume that the perturbation radius satisfies $\delta < R/2$, then there exists a classifier $\hat{f}$ represented by a ReLU net with $\tilde{\mathcal{O}}(\mu N d + \text{poly}(d))$ parameters such that*

- *For adversarial robust training error, $\mathcal{L}_{\mathcal{S}}^{adv,\delta}(\hat{f}) = o(1)$;*

- *For clean test error, $\mathcal{L}_{\mathcal{D}}^{clean}(\hat{f}) = o(1)$;*

- *For adversarial robust test error, $\mathcal{L}_{\mathcal{D}}^{adv,\delta}(\hat{f}) = \Omega\left(\mathcal{L}_{\mathcal{D}}^{adv,\delta}(f_{clean})\right)$.*

However, while the previous results have shown that achieving both low robust training error and high clean test accuracy is representatively efficient for ReLU nets, robust generalization still requires exponentially large models even with our data assumption.

**Theorem 3.5.** *Let $\epsilon \in (0,1)$ be a small constant and $\mathcal{F}_n$ be the set of functions represented by ReLU networks with at most $n$ parameters. There exists a sequence $N_d = \exp(\Omega(d)), d \geq 1$ and a universal constant $C_1 > 0$ such that the following holds: for any $c \in (0,1)$, there exists a underlying data distribution $\mathcal{D}$ that satisfies all above data assumptions and is $\mu_0$-balanced, such that for any $R > 2\delta$ and robust radius $c\epsilon$, we have*

$$\inf\left\{\mathcal{L}_D^{adv,c\epsilon}(f) : f \in \mathcal{F}_{N_d}\right\} \geq C_1 \mu_0.$$

*where $\mu_0$-balanced means that there exists a uniform probability measure $m_0$ on $\mathcal{X}$ and the distribution $\mathcal{D}$ satisfies that $\inf\left\{\frac{\mathcal{D}(E)}{m_0(E)} : E \text{ is Lebesgue measurable and } m_0(E) > 0\right\} \geq \mu_0$.*

In other words, the robust generalization error cannot be lower that a constant $\alpha = C_1 \mu_0$ unless the ReLU network has size larger than $\exp(\Omega(d))$.

Therefore, we get the following representation complexity inequality,

$$\text{Representation Complexity:} \underbrace{\text{Clean Classifier}}_{\text{poly}(d)} < \underbrace{\text{Robust Memorization}}_{\tilde{\mathcal{O}}(\mu N d + \text{poly}(d))} < \underbrace{\text{Robust Classifier}}_{\exp(\Omega(d))}.$$

This inequality shows that, while functions achieving robust memorization have mildly higher representation complexity than clean classifiers, adversarial robustness requires excessively higher complexity, which may lead to adversarial training converges to robust memorization regime.

## 4    ROBUST MEMORIZATION ON REAL IMAGE DATASETS

In this section, we demonstrate that, on real image datasets, adversarial training can learn classifiers in robust memorization regime. Indeed, we need to study whether models trained by adversarial training tend to memorize data points by approximating local robust indicators on training data.

Concretely, for training data $(x, y)$, we use two measures, maximum gradient norm within the neighborhood of training data, $\max_{\|\xi\|_\infty \leq \delta} \|\nabla_x \mathcal{L}(f(x + \xi), y)\|_1$ and maximum loss function value change $\max_{\|\xi\|_\infty \leq \delta}[\mathcal{L}(f(x + \xi), y) - \mathcal{L}(f(x), y)]$. The former measures the $\delta$−local flatness on $(x, y)$, and the latter measures $\delta$−local adversarial robustness on $(x, y)$, which both describe the key information of loss landscape over input.

**Experiment Settings.** In numerical experiments, we mainly focus on two common real-image datasets, MNIST and CIFAR10. During adversarial training, we use cyclical learning rates and

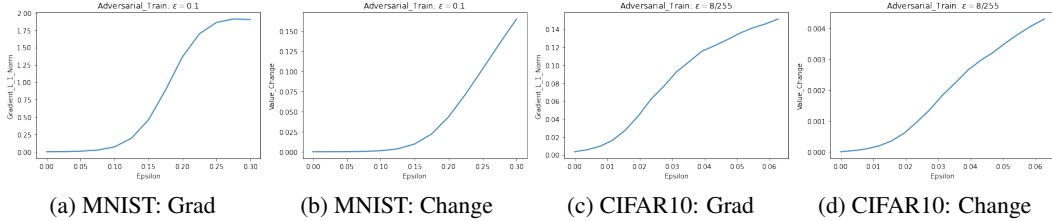

| (a) MNIST: Grad | (b) MNIST: Change | (c) CIFAR10: Grad | (d) CIFAR10: Change |

Figure 1: **(a)(b):** Robust Memorization on MNIST with Training $l_\infty$ Perturbation Radius $\epsilon = 0.1$; **(c)(d):** Robust Memorization on CIFAR10 with Training $l_\infty$ Perturbation Radius $\epsilon = 8/255$.

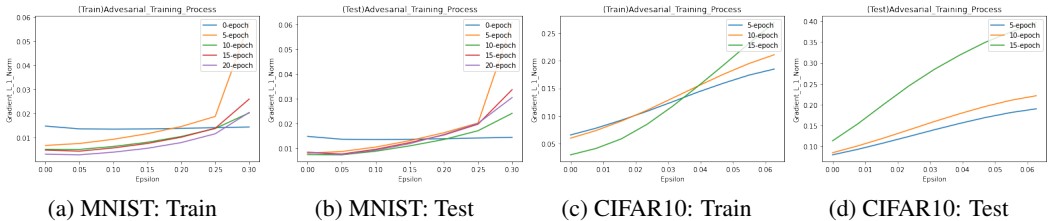

| (a) MNIST: Train | (b) MNIST: Test | (c) CIFAR10: Train | (d) CIFAR10: Test |

Figure 2: **(a)(b):** Learning Process on MNIST with Training $l_\infty$ Perturbation Radius $\epsilon = 0.1$; **(c)(d):** Learning Process on CIFAR10 with Training $l_\infty$ Perturbation Radius $\epsilon = 8/255$.

mixed precision technique (Wong et al., 2020). On MNIST, we use a LeNet5 architecture and train total 20 epochs. On CIFAR10, we use a Resnet9 architecture and train total 15 epochs.

**Robust Memorization.** To verify robust memorization in adversarial training, we first use the standard AT to train models by a fixed perturbation radius $\epsilon_0$, and then we compute empirical average of maximum gradient norm and maximum loss change on training data within different perturbation radius $\epsilon$. We can see numerical results in Figure 1, and it shows that loss landscape has flatness within the training radius, but is very sharp outside, which practically demonstrates robust memorization on real image datasets, including MNIST and CIFAR10.

**Learning Process.** We also focus on the dynamics of loss landscape over input during the adversarial learning process. Thus, we compute empirical average of maximum gradient norm within different perturbation radius $\epsilon$ and in different training epochs. The numerical results are plotted in Figure 2. On both MNIST and CIFAR10, with epochs increasing, it is observed that the training curve descents within training perturbation radius, which implies models learn the local robust indicators to robustly memorize training data. However, while the test curve of MNIST has the similar behavior to training, on CIFAR10, the test curve ascents within training radius instead, which potentially explains why robust generalization gap on CIFAR10 is more significant than that on MNIST.

## 5 THEORETICAL ANALYSIS OF ADVERSARIAL TRAINING ON LINEAR SEPARABLE DATA

In this section, we theoretically demonstrate robust memorization by analyzing the convergence of adversarial training on a certain synthetic dataset.

Specifically, we assume that data $x \in \mathbb{R}^d$ can be formalized as $cyw^* + \xi$, where $w^*$ is the target direction ($\|w^*\|_2 = 1$), $y \in \{-1, +1\}$ is the label, $c$ is the norm scale that is $\alpha$ for large margin data and is $\beta$ for small margin data ($\beta \ll \alpha$), and $\xi \sim \mathcal{N}(0, \mathcal{I}_d - w^* w^{*\mathrm{T}})$ is a random Guassian noise orthogonal to $w^*$.

Indeed, this synthetic dataset captures the main characteristics of data assumption that we mention in Section 3. On one hand, there exists a simple but non-robust classifier $f_{\mathrm{clean}}(x) = w^{*\mathrm{T}} x$ that can clean classify data but is non-robust for small margin data. On the other hand, due to the randomness of noise, in the high-dimensional setting, small data is also well-separated.

We consider a mixed learner model as $f(x) = w_0^{\mathrm{T}} x + \sum_{i=1}^m a_i \phi(w_i^{\mathrm{T}} x)$, where the neuron $\phi(t) = \sigma(t - b)$, $\sigma(\cdot)$ is ReLU activation function and $b$ is a threshold.

With $N-$sample training dataset $\mathcal{S} = \mathcal{S}_{\text{small}} \cup \mathcal{S}_{\text{large}}$ i.i.d. drawn from the underlying distribution $\mathcal{D}$, we minimize the exponential loss as $\mathcal{L}_{\mathcal{S}}^{\exp,\text{adv},\delta}(f) = \frac{1}{N} \sum_{i=1}^N \max_{\|\delta_i\|_2 \le \delta} \exp(-y_i f(x_i + \delta_i))$.

By using the standard adversarial training algorithm, FGSM (Goodfellow et al., 2014), we have the following result.

**Theorem 5.1.** *With large margin data $\mathcal{S}_{large}$ as training data, we use FGSM to train the model $f$ when only $w_0$ is activated and zero initialized, deriving a parameter iteration sequence $\{w_0^k\}_{k=1,\dots}$. Then, with high probability over the sampled set, we have $\lim_{k \to \infty} \left\| \frac{w_0^k}{\|w_0^k\|_2} - w^* \right\|_2 = o(1)$.*

Theorem 5.1 shows that, under the linear separable data assumption, high clean test accuracy on large margin data implies good clean performance on small margin data, which can help us understand clean generalization better in adversarial training (see Theorem 6.2 in Section 6).

**Theorem 5.2.** *By using a modified adversarial training algorithm on all training dataset $\mathcal{S}$, we derive a parameter sequence $\{\theta^k = (w_i^k)_{i=0}^m\}_{k=1,\dots}$. Then, with high probability, for $0-1$ adversarial training loss $\mathcal{L}_{\mathcal{S}}^{0-1,adv,\delta}$, it holds that $\lim_{k \to \infty} \mathcal{L}_{\mathcal{S}}^{0-1,adv,\delta}(\theta^k) = 0$.*

In fact, the above theorems show that adversarial training method on linear separable data will converge to the normalized target solution, $f(x) = w^{*\mathrm{T}} x + \sum_{x_i \in \mathcal{S}_{\text{small}}} y_i \phi(\xi_i^{\mathrm{T}} x)$, which is exactly the robust memorization function mentioned in Section 3.

# 6 GENERALIZATION GUARANTEES BASED ON ROBUST MEMORIZATION

Standard generalization bound can not directly explain high clean test accuracy after adversarial training. In general, standard generalization bound can be stated as the following form. With high probability over random sampled training data, we have

$$\mathcal{L}_{\mathcal{D}}^{\text{clean}}(f) \le \mathcal{L}_{\mathcal{S}}^{\text{clean}}(f) + \sqrt{\frac{\text{Complexity}(F)}{N}}.$$

where $N$ is the number of samples, and $\text{Complexity}(F)$ denotes a complexity measure of function family $F$, such as VC dimension and Rademacher complexity.

However, this standard generalization bound can not explain high clean test accuracy after adversarial training. In order to have enough capacity for achieving low robust training error (i.e. $\min_{f \in F} \mathcal{L}_{\mathcal{S}}^{\text{adv},\delta}(f) = 0$), we need to set $\text{Complexity}(F) = O(Nd)$ (due to Corollary 3.4 and the relation between VC dimension and the number of parameters (Bartlett et al., 2019)), which causes the above bound too loose to use.

## 6.1 IMPROVED GENERALIZATION BOUND ANALYSIS BASED ON ROBUST MEMORIZATION

Fortunately, we notice that the robust memorization function $f_{\text{adv}}$ has much lower complexity on all large margin data $(\text{poly}(\text{d}))$ than the complexity on only sampled small large margin data $(O(Nd))$. Inspired by this, we can prove a novel generalization bound by leveraging it.

**Assumption 6.1.** We assume that, for any classifier $f$ outputted by adversarial training under the realistic data assumption that we mention in Section 3, we have $\mathcal{L}_{\mathcal{D}_{\text{small}}}^{\text{clean}}(f) = O\left(\mathcal{L}_{\mathcal{D}_{\text{large}}}^{\text{clean}}(f)\right)$, where we use $\mathcal{D}_{\text{small}}, \mathcal{D}_{\text{large}}$ to denote small margin part and large margin part in the population $\mathcal{D}$.

This assumption has been theoretically verified in Section 5, which means that the clean test accuracy on small margin data can be bounded by the clean test accuracy on large margin data. Indeed, it holds for any homogeneous classifier when we assume that small margin data also has small norm. We leverage this property to prove the following clean generalization bound.

**Theorem 6.2.** *Let $D$ be the underlying distribution that satisfies all assumption in Section 3 and Assumption 6.1. With access to $N-$sample training dataset $\mathcal{S} = \{(x_1, y_1), (x_2, y_2), \dots, (x_N, y_N)\}$ is i.i.d. drawn from $\mathcal{D}$, there exists a modified adversarial training algorithm with perturbation*

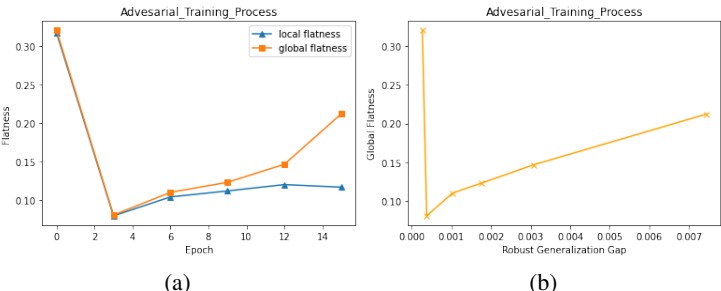

Figure 3: **Left:** Local and Global Flatness During Adversarial Training on CIFAR10; **Right:** The Relation Between Robust Generalization Gap and Global Flatness on CIFAR10.

radius $\delta$, $\mathcal{A}_\delta : \mathcal{D}^N \to \mathcal{F}_\mathcal{S}$, where $\mathcal{F}_\mathcal{S}$ denotes the hypothesis class, such that $\hat{f} = \mathcal{A}_\delta(\mathcal{S})$ satisfies $\mathcal{L}_\mathcal{S}^{adv,\delta}(\hat{f}) = 0$, and it holds with probability high probability that $\mathcal{L}_\mathcal{D}^{clean}(\hat{f}) \lesssim \mathcal{L}_\mathcal{S}^{clean}(\hat{f}) + \sqrt{\frac{\text{poly}(d)}{N}}$.

This result implies that the sample complexity of adversarial training is polynomial in the data dimension $d$, which provides a theoretical guarantee for benign overfitting in adversarial training.

However, we still prove the following theorems to illustrate the hardness of robust generalization. In contrast to the clean generalization, we have an another robust generalization bound that mainly depends on global flatness of loss landscape over input.

**Theorem 6.3.** *Let $D$ be the underlying distribution with a smooth density function, and $N-$sample training dataset $\mathcal{S} = \{(x_1, y_1), (x_2, y_2), \dots, (x_N, y_N)\}$ is i.i.d. drawn from $\mathcal{D}$. Then, it holds with probability $1 - \Delta$ over sampled set $\mathcal{S}$ that,*

$$\mathcal{L}_\mathcal{D}^{adv,\delta}(f) - \mathcal{L}_\mathcal{S}^{adv,\delta}(f) \lesssim N^{-\frac{1}{d+2}} \left( \underbrace{\mathbb{E}_{(x,y)\sim\mathcal{D}} \left[ \max_{\|\xi\|_\infty \leq \delta} \|\nabla_x \mathcal{L}(f(x+\xi), y)\|_1 \right]}_{\text{global flatness}} + \frac{\mathcal{C}(\mathcal{D}, \mathcal{L})}{\sqrt{\Delta}} \right),$$

*where $\mathcal{C}(\mathcal{D}, \mathcal{L})$ is a constant that only depends on the distribution $\mathcal{D}$ and the loss function $\mathcal{L}(\cdot, \cdot)$.*

This robust generalization bound shows that robust generalization gap can be controlled by global flatness of loss landscape over input rather than local flatness. And we also derive the lower bound of robust generalization gap stated as follow.

**Proposition 6.4.** *Let $D$ be the underlying distribution with a smooth density function, then we have*

$$\mathcal{L}_\mathcal{D}^{adv,\delta}(f) - \mathcal{L}_\mathcal{D}^{clean}(f) = \Omega \left( \delta\mathbb{E}_{(x,y)\sim\mathcal{D}} \left[ \|\nabla_x \mathcal{L}(f(x), y)\|_1 \right] \right).$$

Theorem 6.3 and Proposition 6.4 manifest that robust generalization gap is very related to global flatness. However, although adversarial training achieves good local flatness via robust memorization, models lack global flatness, which leads to robust overfitting. This point is also verified by numerical experiment on CIFAR10 (see results in Figure 3). First, global flatness grows much faster than local flatness in practice. Second, with global flatness increasing during training process, it leads to a increase of robust generalization gap.

## 7 CONCLUSION

This paper provides a theoretical understanding of adversarial training by proposing a implicit bias called robust memorization. We first explore the representation complexity of robust memorization under the realistic data assumption. Then, we empirically demonstrate robust memorization on real-image datasets. And we also theoretical analyze adversarial training in the linear separable data setting. Finally, we prove generalization guarantees inspired by robust memorization, which can explain why both good clean performance and robust overfitting happen in adversarial training.

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
