# OpenReview forum: "Towards Understanding Robust Memorization in Adversarial Training"
_ICLR.cc/2023/Conference — Submitted to ICLR 2023_

### Official Review · Reviewer_z1zg · 2022-10-13

**Confidence:** 4
**Correctness:** 3
**Technical Novelty And Significance:** 3
**Empirical Novelty And Significance:** Not applicable
**Recommendation:** 3

**Clarity, Quality, Novelty And Reproducibility:**

The paper needs improvement in its writing to have a better clarity, quality, and reproducibility. From the current presentation, it is difficult to understand the theorems in the paper. If the authors could make the theorems more clear, it could be an interesting paper.

For writing, I would suggest the authors:

[1] Add a brief connection before each theorem and have detailed explanation after the theorem (with intuitions, not math notations).

[2] Be more accurate on the wordings, and try to fix typos.

**Strength And Weaknesses:**

Strength:
It is an interesting idea to connect the adversarial training to some local estimate to explain over-fitting.

Weakness:

Main concern:

[1] The authors need to improve the clarity of the mathematical notations and formulas. For example

(1) Definition of f_adv is confusing. The second part of f_adv is reasonable, but why we take a summation on nearby samples? Is it a summation or an average? Also Sometimes it uses S and sometimes is uses \mathcal{S}.

(2) In Proposition 3.1, the same equation repeats?

(3) In Proposition 3.2, what is "local constant label"?

(4) In Corollary 3.4, if we want to claim that "a neural network with (some number of) weights can achieve robust memorization", should we prove that every network reaching a small robust training error has a poor robust test error? If we have "there exists a classifier", then how can we make sure this is the one the training is converging?

(5) In Theorem 3.5, the notation N_d is used but not defined. There is a definition in Theorem C.1 that N_d is the number of parameters of the nueral network. Please move the definition to the main text.

(6) In Theorem 5.1, how to understand the sentence "...when only w0 is activated and zero initialized, deriving a parameter sequence..."? In Theorem 5.2, again, what is the sequence (k)?

(7) Theorem 6.3 provides an upper bound, and Proposition 6.4 provides a lower bound. Please add some descriptions about these two results in the paper. Also, Theorem 6.2 appears suddenly without any connection.

It is acceptable to have some minor typos, but there are so many typos in the paper and it prevents me from understanding the logic of the paper.

[2] The authors need more evidence to show why their proposed analysis is prefered than other analysis, or why it is essential to consider the robust memorization.

[3] Most papers in the reference list are published before 2021. Could the authors supplement the introduction with some more updated literature?

========================


Minor issues:

[1] About the contribution: is the robust memorization an existing implicit bias in adversarial training? If this is the case, then why the contribution is that "we propose an implicit bias called robust memorization in adversarial training..." in the abstract as well as the introduction? Should it be that the authors figure out a source of the implicit bias in adversarial training and identify it as robust memorization?

[2] Spelling or issue.

(1) Page 2: "is the (samll) margin part of training dataset"

(2) Theorem 5.1, 6.2: "over (the) sampled set", missing "the"

(3) Theorem 6.3: "Let D (is) the underlying", "is" should be "be"

(4) Section 5, paragraph 2: "for small large data"

**Summary Of The Paper:**

This paper studies the robust memorization problem in adversarial training, and explains that why adversarial training suffers from severe over-fitting problem.


**Summary Of The Review:**

The paper has many writing issues. As listed in the weakness part, the big amount of typos/grammars in theorems preventing readers from understanding the paper.

---

> ### Author Response · Authors · 2022-11-18
> **Response to Reviewer z1zg**
>
> We would like to thank you for your careful reading of our paper and insightful comments. We will improve clarity of our paper in the revision.

---

> > ### Author Response · Authors · 2022-11-19
> > **Response to Reviewer z1zg**
> >
> > Dear reviewer,
> >
> > we have updated the revision of our paper.

---

### Official Review · Reviewer_W3fx · 2022-10-18

**Confidence:** 4
**Correctness:** 4
**Technical Novelty And Significance:** 2
**Empirical Novelty And Significance:** Not applicable
**Recommendation:** 3

**Clarity, Quality, Novelty And Reproducibility:**

The paper is written well and clear. But it lacks some necessary references and some important theorems derived in the paper have been proposed in existing works. Therefore I am concerned with the novelty of this paper.

**Strength And Weaknesses:**

Strength:
* This paper thoroughly analyzed the robust memorization in adversarial training, from both representation complexity and sample complexity.
* The paper is well-written and easy to follow.

Weaknesses:
* The paper lacks a large group of necessary references, e.g. [1][2][3][4][6].
* The concept of robust memorization has been introduced and explored by many papers [1] [2] [3].
* The generalization bound of adversarial training has been proved by [6] which is part of the last contribution of this paper.
* In Section 3.2, the paper compares the representation complexity of robust memorization with robust generalization, where it states that the representation complexity of robust generalization is exponential to the dimension. This result is known in [5]. Although the authors have cited this in the introduction, it is not made clear here.

[1] Exploring Memorization in Adversarial Training.

[2] Adversarial Training Can Hurt Generalization.

[3] Towards the Memorization Effect of Neural Networking Adversarial Training

[4] Understanding Robust Overfitting of Adversarial Training and Beyond

[5] Why robust generalizationin deep learning is difficult: Perspective of expressive power

[6] Adversarially Robust Generalization Requires More Data

**Summary Of The Paper:**

This paper theoretically analysed the complexity caused by robust memorization in adversarial training. The paper first proposed the concept of memorization then derived the complexity of neural networks required for robust memorization and for robust generalization. The the paper discussed the sample complexity of adversarial training.

**Summary Of The Review:**

In all, I think the paper is written well and has a clear structure. However, I recommend the authors to include some necessary discussion on their differences and connection between existing works on robust memorization. I listed some related papers in the review but there can be more related papers.

---

> ### Author Response · Authors · 2022-11-18
> **Response to Reviewer W3fx**
>
> We would like to thank you for your careful reading of our paper and insightful comments. We will add some discussion about related work in revision of the paper.

---

> > ### Author Response · Authors · 2022-11-19
> > **Response to Reviewer W3fx**
> >
> > Dear reviewer,
> >
> > we have updated the revision of our paper.

---

### Official Review · Reviewer_59uS · 2022-10-25

**Confidence:** 4
**Clarity, Quality, Novelty And Reproducibility:** 1. Not sure what Figure 1 is trying t…
**Correctness:** 3
**Technical Novelty And Significance:** 3
**Empirical Novelty And Significance:** 2
**Recommendation:** 5

**Strength And Weaknesses:**

Strength
1. Propose a new implicit bias called robust memorization to explain the empirical behavior of adversarial training
2. Provide both empirical and theoretical results showing evidence from different perspectives


Weakness
1. This new implicit bias still remains a conjecture in my opinion. There lacks strong evidence to show that it is actually what happens.
2. The empirical results is somewhat known, and do not provide many new information
3. The authors did not compare their theoretical analysis on robust memorialization with previous theoretical studies in adversarial training


**Summary Of The Paper:**

In this paper, the authors proposed an implicit bias called robust memorization in adversarial training under the realistic data assumption. By function approximation theory, the authors proved that ReLU nets with efficient size have the ability to achieve robust memorization, while robust generalization requires exponentially large models. The authors also demonstrate robust memorization in adversarial training from both empirical and theoretical perspectives.


**Summary Of The Review:**

1. It seems that f_adv in page 2 is constructed, not trained. By “adversarial training error” did the author mean an adversarial loss of training data? If so, why does it implies “global convergence of adversarial training”?

2. This new implicit bias still remains a conjecture in my opinion. First, a network that is able to represent the target functions f_adv does not mean it will lean towards f_adv. Second, the empirical study that the maximum gradient norm is sharp outside the perturbation region only suggests on training data it is robust. It cannot show anything related to the outside data points.

3. Dividing data into small-margin and large-margin ones could fail in modern deep learning situations where each data point can easily find a small margin to any other classes (targeted adversarial examples). In other words, $\mu$ could be just 1.

4. For simple data distributions, previous works have shown that adversarial training can even achieve robust generalization, such as Dan et al. 2020 and

    "Precise statistical analysis of classification accuracies for adversarial training." arXiv preprint arXiv:2010.11213 (2020).

    "Benign Overfitting in Adversarially Robust Linear Classification." arXiv preprint arXiv:2112.15250 (2021).

     Do the authors' results contradictory to theirs? The authors may want to comment on this.

---

> ### Author Response · Authors · 2022-11-18
> **Response to Reviewer 59uS**
>
> We would like to thank you for your careful reading of our paper and insightful comments.
>
> $\textbf{Regarding the constructed function}$ $f_{\text{adv}}$  First, as the reviewer points out, “adversarial training error” refers to an adversarial loss of training data, which is exactly defined as $\frac{1}{N} \sum_{i=1}^{N}  max_{\|\|x'-x_i\|\| \leq \delta} \mathbb{I}${$ \operatorname{sgn}(f(x')) \ne y_i $ } .  Indeed, $f_{\text{adv}} $ achieving zero adversarial training error doesn’t mean that adversarial training algorithms will converge to global optimal solutions. But, it shows the existence of a solution that satisfies all three properties: i) clean test performance is good; ii) robust training error is low; iii) robust test error is high. Motivated by this constructed result, we conjecture that robust overfitting is due to robust memorization. Similar to $f_{\text{adv}}$, functions in robust memorization regime both learn important features for clean data (these features don’t help robust classification), and robustly memorize small margin training data points, leading to both good clean test accuracy and large robust generalization gap. Besides, from the perspective of representation complexity, our constructed result also shows that classifiers in robust memorization regime have much lower representation complexity than true robust classifiers that have a high robust test accuracy, which implies the potential reason why adversarial training converges to robust memorization regime but not a true robust classifier.
>
> $\textbf{Regarding the implicit bias}$ In our work, we empirically and theoretically verify robust memorization in adversarial training. In practice, numerical experiments shows loss landscape has flatness within the training radius, but is very sharp outside, which implies that classifer trained by adversarial training robustly memorizes training data. In theory, we construct a synthetic dataset, under which adversarial training exactly converges to a solution in robust memorization regime during learning process. Thus, robust memorization is not only a conjecture even though now we can not provide a theoretical justification about robust memorization with a general data assumption.
>
> $\textbf{Regarding the small and large margin data}$ First, The existence of small and large margin data for image classification has been empirically verified [1]. Second, if we select an appropriately small perturbation radius in practice, for models trained by clean training, adversarial robust test error will be smaller than 100%, which means that there exists some proportional large margin data that can not be attacked by adversarial attack.
>
> $\textbf{Regarding the simple data distributions}$ In fact, although some previous works theoretically show that adversarial training works on some simple data distribution, even linear separable data doesn’t mean that achieving robust generalization is easy, and robust generalization on linear separable data still requires exponentially large models in worst case [2]. In this paper, we extend results in [2] to a more general setting, where we assume that data can be clean separated by a ReLU network with reasonable size (width and depth).
>
> [1] Banburski, A., De La Torre, F., Pant, N., Shastri, I., & Poggio, T. (2021). Distribution of Classification Margins: Are All Data Equal?. arXiv preprint arXiv:2107.10199.
>
> [2] Li, B., Jin, J., Zhong, H., Hopcroft, J. E., & Wang, L. (2022). Why Robust Generalization in Deep Learning is Difficult: Perspective of Expressive Power. arXiv preprint arXiv:2205.13863.

---

> > ### Author Response · Authors · 2022-11-19
> > **Response to Reviewer 59uS**
> >
> > Dear reviewer,
> >
> > we have updated the revision of our paper.

---

### Decision · Program_Chairs · 2023-01-20

**Decision:**

Reject

**Justification For Why Not Higher Score:**

The paper has limited novelty and did not properly cite/discuss many recent related works.

**Justification For Why Not Lower Score:**

N/A

**Metareview: Summary, Strengths And Weaknesses:**

The paper studies robust memorization in adversarial training. However, the paper did not compare their theoretical analysis and empirical findings with several recent works on robust memorization. Further, there are several other issues with the current submission, including the writing and the clarity of the paper. Therefore we recommend rejection for this paper.